# Organizational Models in European Forestry: An Attempt of Conceptualization and Categorization

Francesco Loreggian *, Laura Secco  and Davide Pettenella 

Department TESAF, University of Padova, Viale dell'Università 16, 35020 Legnaro, PD, Italy
* Correspondence: francesco.loreggian@phd.unipd.it; Tel.: +39-3385002696

**Abstract:** The changes and challenges that are tackling the forest sector in recent decades have prompted governments and foresters to work hard to find innovative solutions. Research in the forestry domain has focused on product and process innovation, and more recently on business systems and social innovation. In addition, organizational innovation is recognized and documented. However, while consistent conceptualization work has been conducted for business models and social innovation, the organizational domain in forestry seems less clear, characterized by multiple actors and often overlapping in literature, while a clear framework to describe it is missing. This work proposes a conceptualization of the "organizational model", a concept embracing different approaches to build an analytical framework used to describe and characterize organizations in the forestry sector. The framework is drafted referring to existing theories, then tested (and further developed) through a semi-systematic literature review on organizations operating in forest management in Europe that are identified, categorized, and characterized. This exercise confirms that forest management organizations can be described with several diverse arrangements and can be complex entities: a holistic and comprehensive approach is more likely to be used by policy initiatives addressing improvement of forest management.

**Keywords:** organizational model; organizational innovation; forest governance



## 1. Introduction

In Europe, in the last three decades, the forest sector has faced economic and societal challenges resulting from many concurrent phenomena: fragmentation and abandonment of forest properties, changes in the ownership structure due to restitution processes in former socialist countries [1–3], the need for reforestation and to improve nature conservation and the provision of forest ecosystem services [4,5], the challenges of climate change [6], and the growing biomass demand connected to the bioeconomy development [7,8]. Moreover, the forest sector is also hit by the most recent global political crisis and its consequences on the market, characterized by instable prices and an increase in energy costs. Changes that occurred in recent decades and increasing market competition are important drivers for forest-related companies to seek new forms of competitive advantage or business opportunities [9], especially when challenged by the increasing societal demand for environmental protection and services. From another (but convergent) perspective, the role of forest management is important for rural and regional development, because forests are often found in disadvantaged rural areas where the use of forest resources represents a significant socio-economic opportunity [10] and multifunctional forest management is recognized as the most practical means for increasing the forest-related ecosystem services, a request coming also from densely populated regions [11].

All these reasons have prompted policymakers (governments) and practitioners (forest management companies) to look for innovative solutions and to develop new business opportunities, increasing their organization's performance [12] and their competitive advantages [8,13]. Various types of innovation have been tackled, from products to process,

marketing, organizational as well as institutional- and governance-related [14–16]. Recently, "social innovation" has been recognized and investigated in relation to forest resources. Social innovation in the forest-related bioeconomy actually includes products, processes, and organizational innovation, which typically also includes social and societal outcomes, while being pushed by initiatives to address social issues [7,14,17].

Considerable research has been conducted, in the forest domain, on product and process innovation and innovation strategy, but less has been conducted on business systems innovation [9,11], i.e., on new ways of managing a business, including the creation of new business models [18]. Business model canvas [19] is a very diffused framework used to represent, evaluate, and design business models and was also applied to analyze forest-based businesses [9]. Assuming that this approach can offer reliable solutions to embrace new business goals and aspects that are not only economic, but also related to the social and environmental dimensions [9], it has been pointed out that many other dimensions are relevant, such as internal values and motivation, governance processes, ownership and legal forms, attitudes and competences, communication, etc. [7,9]. This spectrum is much wider than the one represented by what is commonly called "business model", embracing the legal framework and decision-making processes, the characteristics and values of internal and external actors and their relationships, and the overall organizational arrangement of all these aspects together. This approach is also known as "business model thinking", and it is recognized as a good tool 'to explore the potential of business innovation' [1] (p. 155). However, also in the forest management domain, this approach mainly applies to traditional entities conducting forest-related business, e.g., logging companies, forest management enterprises, and wood-chain brokers. It does not provide a definition seeming to adequately fit all the new "organizational arrangements" (e.g., more oriented toward public–private partnerships, more flexible in adapting to constantly changing scenarios, more interactive with the needs of civil society needs, sometimes based on informal relations and shared values instead than formalized contractual agreements) that are likely needed to successfully deal with the sustainable management of forests in the perspective of current crises and future challenges. Several inconsistent names can be found to define forest management organizations, focusing on specific perspectives such as forest ownership, role in forest management, and legal organizational type; however, a broad framework to catch the complexity of the organizational domain is missing.

To adequately support innovation in this field and to design and implement useful policy tools, it seems useful to try to clarify the meaning, perimeters, and key features of what could be considered an "organizational model" in the forest business area. Since the scientific research on this topic is fragmented and does not provide a comprehensive and updated conceptualization, the main goal of this paper is to propose an attempt of comprehensive conceptualization that is suitable to draw a characterization of forest management organizations.

With this general goal, this work is based on two main subobjectives: (i) defining an analytical framework that can be used to describe and analyze various types of forest-specific organizational models; and (ii) testing the analytical framework on existing organizational arrangements within forest management organizations in Europe. Results, elaborated into recommendations, are ultimately intended to provide support to policy makers, in the definition of financial and regulatory instruments addressing the purpose of innovating the forestry sector, as well as to practitioners/forest managers and companies, especially in identifying possible areas of improvement and innovative solutions to institutional requirements, financial constraints, or internal blocks.

## 2. Approaches, Materials, and Methods

### 2.1. Guiding Approaches, Framing, and Concepts

Starting from a semantic definition, in the Cambridge dictionary, https://dictionary.cambridge.org/dictionary/english/organizational (accessed on 20 April 2023), the adjective 'organizational' can be related to the ability to plan, to belonging to a group (organization),

to the combination of a system to make it work; in any case, it is relative to the verb 'to organize', as for the case of the noun 'organization': 'a group of people who work together in an organized way for a shared purpose'. A model, https://dictionary.cambridge.org/dictionary/english/model (accessed on 20 April 2023), can be more easily defined as 'a representation, a simple description of a system or process'. Combining these two definitions, to characterize an organizational model means answering the following questions: (1) who (are the members of the organizations, i.e., the components of the group of people who work together?) (2) what (does the organization do?); (3) how (does the work run? i.e., how is the "organized way" of doing the work structured?) and (4) why (do the members work together, i.e., what is the shared purpose?).

Constitutive principles of organizations can be found in diverse fields of study, such as: (i) economy, law, and business management, (ii) social sciences, and (iii) policy sciences. In the following, some key features characterizing organizations and organizational settings are picked up from these fields and are finally combined together to frame a general concept of 'organizational model', which we propose to analyze and cluster forest management organizations, as reported in Section 3.

### 2.1.1. Economy, Law, and Business Management

The organization as a firm based on "nexus of contracts" was introduced in the 1970s by some economists and became a pillar for analyzing organizations as entities defined by law [20,21]. This concept describes the way two or more persons coordinate their economic activities by saying that a common approach is that each of these persons enters into a contract with a third party, called a "firm" (i.e., a formal organization), who undertakes the coordination through design of the separate contracts and, most importantly, through exercise of the discretion given to the third party by those contracts. Productive activity is commonly organized in the form of large nexuses of contracts [22]. A firm must generally have two basic legal attributes: well-defined decision-making authority and the ability to bond its contracts credibly, by means of a pool of assets that the firm itself or the firm's managers can offer as satisfaction for the firm's obligations toward creditors, while securing the firm itself (its assets) with respect to the personal obligations [23].

In business sciences, business models were developed to describe what and how an (economic) organization does. They are defined as a representation of the underlying core logic and strategic choices to create and capture value within a value network [24]. The business model influences (and derives from) organizational choices, which include "the value a company offers to one or several segments of customers and the architecture of the firm and its network of partners to create marketing and delivering this value and relationship capital to generate profitable and sustainable revenue streams" [25]. Therefore, it can be considered part of the whole, with respect to the broader concept of the 'organizational model'. The variables describe what an organization does (the value proposition), who is addressing whom (the clients and beneficiaries), and by which means (the key resources and activities).

### 2.1.2. Social Sciences

Richard Scott [26] proposed a classification of organizational theories into three categories: rational, natural, and open systems approaches. According to his categorization, organizations as rational systems are oriented to the pursuit of specific efficiency goals and exhibit highly formalized structures; the natural systems approach considers organizations as interpretation systems that scan, interpret, and learn while acting in mutual dependencies with their social environment; the open systems approach proposes an understanding of organizations as deeply socially embedded, shaped, supported, and infiltrated by their environments. A group of sociologists focused on the idea that organizations are the result of decisions [27] and are a social order that is intrinsically dynamic and could be contrasted with more static orders, such as institutions and networks [28]. They identified five fundamental decisions that determine organized social interaction: (i) decisions on membership

define who is a member of the organization and who is not; (ii) decisions on rules regulate what the members must do and how to do it; (iii) decisions on monitoring allow the participants to observe each other, to control but definitively to know how to operate; (iv) decisions about sanctions (positive and negative) are set to enforce other decisions; and (v) decisions on hierarchy establish who has the initiative and power (for decision making). Furthermore, if a 'partial organization' can exist [29], organizational features can also exist outside the context of formal organizations, when only some of the five fundamental decision levels are (eventually partially) implemented. Adopting a broader institutional lens, organizations are recognized in sociology as basic institutions, with institutions being foundations that make up the social life, "the prescriptions that humans use to organize all forms of repetitive and structured interactions" [30], which can be formal or informal, as "socially shared rules, usually unwritten, that are created, communicated, and enforced outside of officially sanctioned channels" [31]. This allows one to also consider informal and not fully structured organizations and organizational models that can be frequently found in the forest domain [32,33], as relevant for policies and practices. Two more key concepts can be found in the neo-institutional perspective: organizations configure and reconfigure their structures and practices to demonstrate alignment with the goals and values expected within their institutional environment and to gain legitimacy from other actors; isomorphism occurs in organizations when addressing institutional change, through coercive, mimetic, and normative processes [34]. The social (and ecological) context gains much relevance under these perspectives, as the consideration of organizations as dynamic entities whose arrangements change, as the results of internal decisions, to adapt to their environment, sensitive to conflicts, and to social consensus.

2.1.3. Policy Sciences

Within the rich literature about policy sciences, a specific approach proposed to describe policy arrangements was acknowledged as particularly inspiring, suggesting a synthetic framework suitable to frame the set of features selected from other disciplinary domains. According to the work of Wiering et al., a "policy arrangement" is defined as "the temporary stabilization of the content and organization of a policy domain" [35], to describe the way a certain policy domain is (temporarily) shaped in terms of organization and substance. On the other hand, institutionalization incorporates the development of structures as a result of actions and behaviors that, in the search for stabilization, in turn are subject to continual change and adjustment [36]. In these scholars' work, four analytical dimensions are proposed to understand policy design and practices: discourses, power, rules, and actors, which are inextricably interwoven [37]. Actors are those who are involved in the policy process, whose power refers to the mobilization and deployment of resources. Rules of the game describe both laws regulating the policy domain and formal or informal procedures for decision making, while discourses refer to ideas, values, views, and narratives of the actors involved. A change to a temporary policy arrangement can result from a change in any of the above-mentioned dimensions, therefore setting up a new stabilization. Turning this approach to organizations, while keeping an institutional perspective, means to accept the idea that it can result as the development of structures from people's (actors') choices and behavior that stabilize and change as soon as any of its key dimension changes. In fact, organizations also result as the development and implementation of "rules" to allow a group of "actors", given a set of "resources and power", to achieve their "purpose", according to a system of values ("discourses") [35,36], being the key dimensions highlighted by the "inverted commas".

Another interesting contribution to the concept of the organizational model can be found in Krott's actor-centered power (ACP) approach: given the recognition of actors and power as key dimensions of our model, it seems appropriate to consider the definition of power as the "capability of an actor to influence other actors" [38]. Evidently, this is very relevant in an organizational arrangement, where actors have a central role and power can determine whether many choices are to be implemented or not. Though deepening

the theoretical roots of ACP theory is not the scope of this work, the three elements forming the social relation called actor-centered power are recognized as key features of the organizational model's conceptualization. This means that in analyzing organizations, the assessment of power should be based on the recognition of the ability of actors to apply the strategies of coercion, incentives, and dominant information. This assessment was already applied to forest governance to understand how power shifts in governance can influence actors' power relations with respect to their interests in forest ecosystem services [39]. The scale of the organizational model is something different from Krott's application to forest policy and governance; nevertheless, power dynamics are hypothesized to be very similar in an organizational context, within and between organizational entities.

### 2.1.4. Framing Variables

Trying to merge these approaches, organizations can be seen as institutions that are subject to a continue dynamism between stabilization and change, in search of adaptation and innovative solutions to continuously emerging challenges and opportunities, sensitive to their social and ecological context. Inspired by the definition and variables of the policy arrangement approach (PAA) [36] definition and variables, we believe that an "organizational arrangement" could be defined as the temporary stabilization of the content and organization of an organizational domain.

The key dimensions of the PAA framework, slightly revisited, are suitable to group organizational features resulting from the multidisciplinary approach adopted in this first part of the research. The key dimension "actors" addresses the question "who (are the members)"; the key dimensions "rules" and "power and resources" describe "how (the organization works)"; while the dimension "discourses" is suitable to display "what the organization does" and why. Furthermore, organizations are settled in a context (frequently) cited as "(social) environment" within the neoinstitutional literature.

Twenty basic characteristics (or variables) were identified within the aforementioned approaches to characterize organizations, and—inspired by the PAA's framing—they were grouped into four (plus one) key dimensions: (i) actors, (ii) values and discourses, (iii) rules, and (iv) power and resources, embedded into a context. This latter is considered a fifth key dimension. These four (plus one) key dimensions constitute upper-level categories and were used to draw an attempt of a comprehensive analytical framework, represented in Figure 1, where 20 key features are represented as belonging to one or more key dimensions.

### 2.2. Research Plan, Materials, and Methods

At the beginning of this research, the need to clarify the key concept of 'organizational model' was identified, which encompasses in some way all the key characteristics that portray an organizational arrangement, which is not intended as a static form, but rather a combination of dynamic variables.

As displayed in Section 2.1, concepts from different theories and disciplinary domains were selected to elaborate our conceptual framework, not necessarily exposing and confronting whole theories nor exploring all the possible approaches to the topic from all the disciplines. Rather, the identification of parts of theories mentioned within the literature reviewed led to a selection of suitable theoretical contributions. This conceptualization was detailed with 20 key characteristics grouped into 4 (plus 1) key dimensions, represented in Figure 1, to compose the analytical framework that was applied for the analysis of forest management organizations within the scientific literature. The characterization of such organizations is displayed according to this framework in Section 3, where organizational models detected are clustered into 6 categories.

In the end, the analysis of the literature review addresses two goals at the same time: the characterization and categorization of forest management organizations, while testing (and improving) the conceptual framework hypothesized in the first part of this research, which also means to challenge and improve the overall conceptualization of the

'organizational model'. The methodology for the literature review and content analysis is described below.

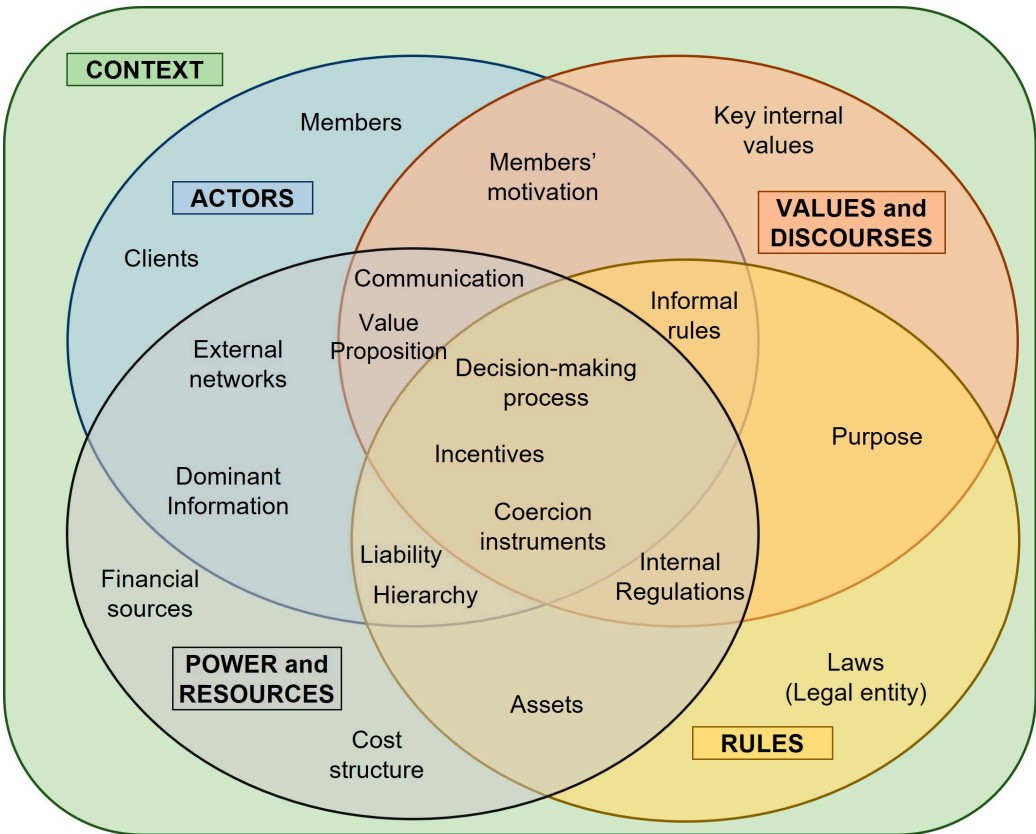

**Figure 1.** Conceptual and analytical framework for the study of organizational models, inspired by and based on the PAA framework [36,37]. Source: own elaboration.

### 2.2.1. Review of the Literature

A semi-systematic review of the literature was conducted, focusing on organizations and organizational issues in the forest sector. Designed for broad topics that have been studied by various disciplines [40], this methodology is suitable for providing an understanding of complex areas, while being transparent and allowing readers to assess whether the arguments for the judgments made were reasonable, both for the chosen topic and methodology [41]. The main steps of this methodology were applied: (1) identification of studies to be included, (2) screening of identified studies, (3) eligibility assessment, (4) full document reading, and (5) data extraction [42].

This review looks at forest management organizations in Europe, considering the geographic area. Between June and October 2022, 29 query strings were entered on the scientific database Scopus, with an iterative approach through 4 stepwise blocks of searches. The four blocks were built of strings based upon two keywords: "Forest AND organi?ation" plus one or more keywords added using Boolean operators such as W/1, W/2, AND, or OR, chosen within four categories: organizational sciences' key topics (block 1); synonymous locutions close to the concept of 'organizational model' (block 2); types of formal organizations (block 3); synonymous with 'collaboration' (block 4). The words 'timber' or 'wood' were excluded from the keywords' selection, as they would have produced results about the industrial timber transformation chain rather than forest management. The words 'organization' and 'organisation' were both considered, using the '?' character in the query strings. The general strategy of the review process is represented in Figure 2, while a complete list of the query strings applied is detailed in Table S1, at the end of the manuscript, where the number of articles selected within each search is also reported. The

whole process represented in Figure 2 was carried out per each block of searches, then was reiterated 4 times (one for each block). After reading the selected articles resulting from each block, the keywords to compose the following block of queries were defined until 4 blocks were completed.

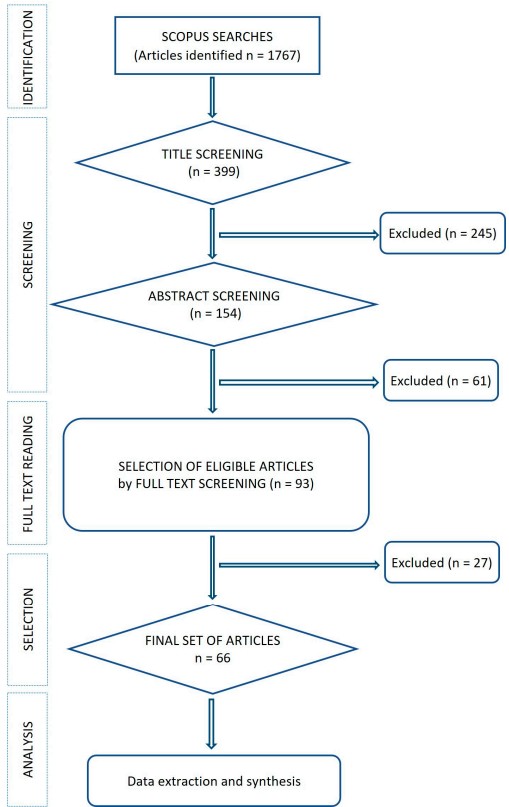

**Figure 2.** Diagram of the research process for the semi-systematic literature review. Source: own elaboration, based on PRISMA flow diagram, adapted from Page et al. [42].

All searches were filtered by subject area, limiting to agricultural and biological sciences, environmental sciences, earth and planetary sciences, social sciences, business management and accounting, decision sciences, engineering, energy, and economics. The results were also sorted by language, selecting only English, and by type of document, limiting only to articles, reports, and reviews.

An initial list of 1767 articles were retrieved in total. After selecting them on the basis of title (only titles explicitly indicating forest-related topics and revealing a reference to organizations or organizational topics were selected), 399 articles were further analyzed. After reading the abstracts, 154 articles were selected. Among these, 28 duplicates were deleted, while 33 were excluded after reading the full text, because they were not consistent with the objectives of the research. Therefore, a list of 93 papers was selected for full reading, and 27 more were excluded because they were not fully consistent or relevant for the purpose of the research, finally resulting in 66 articles to compose the ultimate set of this review of the literature. The detailed list of selected articles is presented in Table S2 in the Supplementary materials.

### 2.2.2. Content Analysis

The content analysis for data extraction from the ultimate set of articles was carried out applying a meta-ethnographic approach, a method developed to establish a new theory or synthesis and to explain the range of research findings encountered [43], which is applicable to literature reviews, too. Thanks to this inductive approach, the textual content of published studies (rather than the original data of each) was reanalyzed and compared,

first to produce a synthesis of the overall "organizational model" concept suitable for the forestry sector and, secondly, to identify and characterize a selection of categories for forest management organizations displaying similar characteristics. Through meta-ethnography, separate parts are brought together to form a 'whole', so that the result is greater than the sum of its parts [43,44]. Following this approach, while reading the selected articles, all data regarding the 20 organizational models' features, as conceptualized and represented in Figure 1, were identified and copied into a matrix. The matrix was framed by writing in column headers the names of the organizational types described within the articles, while in rows, the concept of the 20 variables composing the organizational model concept (see Figure 1) were described.

While reading and extracting data, two main criteria were recognized as those most frequently determining the definition of an organizational type: i) who the members are and ii) who the forest owners are, the two being either coincident or distinct. These two variables were recognized as the most independent in the framework. Therefore, all the organizational types cited in the articles were clustered until six ultimate categories of forest management organizations were identified according to those two main criteria. Some more typologies resulted, as displayed in Section 3.6 dedicated to "Other organizations", some of which are surely relevant from a practical and political point of view; nevertheless, they were significantly under-represented in the literature review with respect to others, being cited in not more than two articles. The choice not to complete the full analysis for these organizations was made because only some of the 20 variables of the analytical framework could have been described (as reported in Section 3.6) based on data gathered through the selected methodology. Indeed, to complete the description of all (or nearly all) the 20 features, data were synthesized from all articles where organizations belonging to that category are cited to obtain the most comprehensive characterization per each of the six categories.

## 3. Results

Based on locutions and definitions found in the literature, similar organizational models applied to forest management activities in Europe were grouped into six categories, plus a miscellanea one (see Table 1), named accordingly, and characterized by the main organizational traits emerging from the analysis, based on the framework represented in Figure 1. In several cases, similar types are cited in the literature with different names, according to the country-specific legal framework, to different literature streams, or to different perspectives toward similar subjects. For example, "forestry contractors" and "forest enterprises" are used in different articles to indicate the same subject, also cited as "forest harvesters", and were grouped into a unique category, while in the case of "community forests", "community forestry", and "community-based forest enterprises", a partial overlapping of concepts required a more refined distinction and definition.

**Table 1.** Organizational models aggregated and described.

| Organizational Models as Defined by Authors in the 66 Articles | Aggregative Name Proposed | Description |
|---|---|---|
| State forest management organizations<br>State-owned enterprises<br>State forest organizations | State forest management organizations (SFMOs) | A state-owned forest company, enterprise, or agency that performs sustainable forest management and wood production as its major concern; they meet both social and financial objectives, while protecting forests and biodiversity [5,45]. |
| Forest owners' associations<br>Forest owners' organization<br>Forest owners' cooperation<br>Organization of forest owners<br>Forest groups | Private forest owners' organizations (PFOOs) | Private forest owners constitute members' controlled organizations with the aim to represent the interests of the members and/or providing forest management services to optimize management costs and overcome issues due to land fragmentation [46,47]. |

Table 1. *Cont.*

| Organizational Models as Defined by Authors in the 66 Articles | Aggregative Name Proposed | Description |
|---|---|---|
| Common property organization<br>Community forests<br>"Consorzi vicinali" | Community forests (CFs) | Organizations operating forest management based on common ownership rights, management, and use of forests [48]. |
| Community forestry<br>Community forestry enterprises<br>Community-based forest enterprises | Community forestry and Community-based forest enterprises (CBFEs) | In community forestry, communities that do not own forests have some involvement in forest management, decision making, and/or governance and gain some benefit from them [49]. CBFEs are companies organized by community members to actively provide forest products and services, with the goal of producing social returns and/or managing assets that benefit those communities [50]. |
| Social enterprises<br>Not-for-profit enterprises<br>(Rural) Charities<br>Third-sector organizations | Social forest enterprises (SFEs) | SFEs are companies not acting for profit but are established for social or/and environmental purposes [51]. They can be established within a community (forest), but do not necessarily involve forest owners as members. |
| Forest harvesting entrepreneurs<br>Forestry contractors<br>Forest enterprises<br>Forest cooperatives | Forest enterprises (FEs) | Organizations whose business is based upon forest operations, contracted with public or private forest owners (our elaboration), normally not holding forest planning responsibilities and not owning forest land. |
| Environmental organizations<br>ENGOs<br>Certification schemes<br>Certification groups<br>Model forests | Others | Several more types of organizations were mentioned within the 66 articles, but not described with sufficient data to create a category and carry out the characterization analysis. Results are reported in Section 3.6 |

Source: own elaboration.

A bit less than 80% of the articles focus on specific types of organizations, while only about 20% discuss organizational aspects in general terms, and in just a couple of articles, the 'organizational' topic explicitly referred to a clear theoretical framework, as in the case of the "business model canvas" applied to analyze forest-related business models in Europe [9]. "Organizational models" were conceptually framed and described to characterize the organization of private forest owners (PFO) in Austria [52] and adopted as a framework in a study on PFOs' capacity to increase wood mobilization In Slovenia and Serbia [53]. While the business model canvas is designed to outline the arrangement of a business with that perspective, with scarce attention to the organization's members and to their organizational arrangement, the four organizational models proposed for the analysis of PFOs' organizations are based on members' participation in decision-making, management, and on profit allocation. In some articles, "organizational models" were intended to represent the organization of specific aspects, either governance arrangements in the establishment of systems for the payment of forest environmental services [54] or marketing strategies for the commercialization of non-wood forest products [55]. In the 66 articles reviewed, the concept of "organizational model" (or similar locutions) was never applied to achieve a complete representation of the organizational arrangement, as hypothesized in Section 2.

Only 8 articles of 66 are not country-specific, while 58 refer to one or more than one country's cases. A first group of articles (n = 9) refers to countries of Central and Eastern Europe, such as Estonia, Lithuania, Latvia, Poland, the Czech Republic, Slovakia, and Romania; another group is focused on Balkan countries (n = 10); and British (n = 5) and Fennoscandian (n = 15) countries are also well represented, while other European countries are less represented, with only one article dedicated respectively to France, Belgium, Germany, Italy, Austria, and Switzerland, with some more being referred to larger regions, to the whole continent, or to the topic in more general/global terms. The following subsections describe the results in detail, focusing on the four key organizational

dimensions that guided our analysis (i.e., actors, discourses, rules, power, and resources). This description is complemented by Table S3 in the Supplementary Materials.

### 3.1. State Forest Management Organizations

Public forests in Europe account for nearly 40% of the total woodland area, with a quite diverse distribution among countries, from less than 25% in Austria, Norway, France, and Slovenia to more than 70% in Croatia, Czech Republic, and Poland, only to cite some examples [56]. Despite these differences, state forest management organizations (SFMOs) have traditionally played a major role in the forest sector in European countries. The state or local public authorities manage their forests through state-owned forest companies, eventually entrusted with public authority [57]. In the literature, such organizations were reported within eight articles with several different (even if similar) names (see Table 1). The lack of a common terminology could have impeded a deeper understanding of the key role of these organizations in forest management at the regional level.

(A) Actors. The state and its decentralized regional or local authorities are the owners of SFMOs. Evidently, significant differences can be found between countries depending on the organization of the public administration. Forest management can be either assigned to a unique large enterprise, managing all of the state forests, as in the case of Poland, Serbia, and France, or be shared between many smaller local enterprises owned by the regions or the municipalities, as we can observe in Lithuania, Spain, and Italy [45,58]. The smaller the administration (and the forest), the greater the need to optimize the costs of management; therefore, some Italian municipalities, for example, aggregate in forest consortia, which can also include private owners amongst their members [59]. SFMOs sell their forest products and services to other actors of the value chain, such as timber companies and sawmills. They changed significantly in eastern Europe from the 1990s following a wave of privatization and the simultaneous collapse of socialist regimes, induced privatization of the forest industry, the formation of a free timber market with increasing timber imports and exports, as well as new modes of ownership and enterprises [58]. However, SFMOs remain protagonists in European forest management and are almost all represented under the umbrella of EUSTAFOR, an important second-tier organization whose members provide employment to more than 100,000 people; its main goal is to support and strengthen state forest management organizations in Europe, helping them to maintain and improve their economically viable, socially beneficial, culturally valuable, and ecologically responsible practices [5].

(B) Values and Discourses. The state exercises ownership over its enterprises in the interests of the public. The main purpose of state ownership should be to maximize value for society through efficient use of resources [60]. In European forestry, sustainable forest management has provided the guiding principles for SFMOs since the 1990s, and also, more recent concepts such as ecosystem services have forced them to rethink their management goals and to orientate toward a full integration of their social, economic, and ecological dimensions. Therefore, the purpose is to maintain the main function of production (and economic viability) while guaranteeing the provision of ecosystem services of public utility, such as sequestration of C, biodiversity conservation, landscape maintenance, recreation, and soil and water protection [5,45,54,59]. SFMOs can provide forest-related services to private forest owners. In Slovakia and Estonia, SFMOs also manage woodland for absent private owners, and they actively develop new business activities; among the most common are sources of renewable energy, real estate, and recreation activities, but they also develop forest/environmental education, manage forest museums, and nature centers [45,58]. Furthermore, the implementation of the communication process, through education or pure communication campaigns, allows for the reducing of social conflicts and achieving of better compromises in an attempt to find the right balance between production goals and social/environmental purposes [5].

I Rules. SFMOs can assume different legal status, state-owned joint stock companies, pure state enterprises, or other types of profit-making companies. The way SFMOs are

organized and managed is often predetermined by the specific conditions of the forest sector in the country; in general terms, their internal governance is typically hierarchical and functions as a private unit, where decision making is often influenced by political power [45,58] either to lead more commercial-oriented organizations or to provide specific ecosystem services of public interest. In any case, many organizations must integrate all these goals into their development and all SFMOs must follow the rules of sustainable forest management [45]. In Serbia, the SFMOs 'Srbijasume' and 'Vojvodinasume' give professional and advisory support to private owners to enforce sustainable forest management, according to a law enforced in 2011, if the organizations do not directly employ licensed forest engineer [61].

(D) Power and Resources. Competences and powers over forests are often separated with dedicated agencies for state forest administration, law enforcement, and management enterprises. SFMOs operate on the principle of financial self-sufficiency and cover their costs with their own revenues, with a positive financial result [45]. Forest management is held internally by bigger SFMOs, whereas smaller organizations involve external forest consultants. In any case, rationalization and privatization processes, often under pressure for public funding cuts, push the transfer of many forest operations to private contractors, from harvesting to transport and, less frequently, to forest protection services [58,59]. Increasing outsourcing of activities corresponds to a consequent reduction of SFMO personnel [45]. Many SFMOs stipulate long-term contracts with loggers or timber companies, which participate in tenders, while in the case of smaller SMFOs, logs are sold in smaller quantities through auctions. In countries where there is only one large SFMO, the state's role in stabilizing the local timber market is evident, especially during economic crisis or natural disasters; contrarily, in countries where SFMOs are many and smaller, their (low) power is limited to their own forest land and resources [58,59].

*3.2. Private Forest Owners' Organizations*

The PFOOs are the protagonists of 18 articles. Private land fragmentation, along with the lack of organization and insufficient motivation of private owners for harvesting, are cited as some of the most important problems affecting the forest sector in many European countries [62]. The Confederation of European Forest Owners, the European umbrella organization of major national private forest owner associations, advocates the practice of joining cooperatives or associated organizations for forest owners as a good and efficient tool to mobilize the management of unmanaged private forest resources, enabling owners to be well-informed and actively participate in the wood market, while providing a reliable source for the representation of members' interests [46]. These cooperation-based organizations are highly dependent on membership growth [46] and are often encouraged by governments and promoted by foresters [60]. Increasing the participation rate of private forest owners is important to address the long-term requirements of the market and to fulfil effective representation of interests [63].

(A) Actors. Nonindustrial private forest owners are the main actor of this typology. In Europe, 56% of the total forest area is private, of which almost 77% is owned by "individuals and families", while an even higher share of the holdings, 88%, is smaller than 10 hectares [56]. Private forest owners' organizations are diffused in most European countries, with relevant differences. In the Fennoscandian countries, PFOOs have a long tradition: in Sweden, many organizations were founded in the early twentieth century between family forest owners, which own almost half (48%) of the Swedish forest land [64]; they follow cooperative principles of member ownership [65], and in 2013, they handled 50% of the volume cut by family forest owners, corresponding to approximately 25% of Sweden's annual cutting rate [66]. In Norway, already in the early 2000s, 3/4 of timber sales were made by associations of forest owners associations [67], while some Finnish cooperatives, whose members are small private forest owners, are today among the largest forest companies in the world [68]. In these models, owners are members but act much more like shareholders; rather than directly participating in management activities, they

could even be completely absent, thus establishing pure 'dividend' models where they are involved only to make very general decisions and to share profits [52]. When adopting a more 'cooperative' model, PFOOs are associations of active owners directly involved in forest management and operations, while the organization sells assortments, completes contracts, and sorts invoices [53,69]. The French CNPF—Centre National de la Propriété Forestière—is a singular case that is worth mentioning, being a central public institution with 11 regional delegations, grouping approximately 3.5 million private forest owners, thus revealing almost all of the 75% of the French private forests, with some of them being members of PFOOs while others are not [2]. Very small and fragmented forest properties, which cannot offer significant economic benefits and characterize many European countries, currently represent one of the reasons for establishing organizations but also a limit for owners to be interested in joining. Some research has shown that PFOOs cannot be established or succeed everywhere, as reported in some studies in the Balkan area or in the Baltic republics [61,70]. Not all joint activities, knowledge exchange, and cooperation in general must take place in a particular organizational form. Nonmembers of an organization still might successfully cooperate with other forest owners despite their individualistic approach and indecision toward associations, eventually purchasing services from PFOOs. Un upper level is represented by "umbrella organizations", which are larger organizations whose members are PFOOs, such as the Confederation of European Forest Owners (CEPF), at the European level, but several others exist at the national level: they are an important player in the external network of PFOOs to achieve one of their main goals, which is the representation of the interests of members in policy advising [53].

(B) Values and Discourses. According to some works [46,47,71], PFOOs can be divided into two main typologies based on their main purpose: organizations focused on gaining political influence and organizations aimed at improving management, logistics, marketing, and general technical and administrative support. PFOOs often start to achieve one of those two main goals, but after some time they often encompass both, after they grow up, as for the case of PFOOs in the Balkans and in Baltic republics, where they were first intended to give the opportunity to forest owners to be represented in the land restitution process occurring in these countries since 1990 and then gained more competency and importance not only in influencing forest policymaking, but also in offering services to their members [47,61,72]. Moreover, PFOOs whose mission is to provide services and commercial opportunities to owners who are members can also implement their business strategy and sell the same services to other nonmember owners. In general terms, PFOOs can succeed if they have clear objectives to attract members and produce benefits for existing members by reducing the membership costs via doing so [46]. However, many forest owners are described as still reluctant to join such organizations, despite cooperation being encouraged by policy to enforce the sustainable management of private forests, yet only a small share of private forest owners joined an association. Their resistance seems to be mainly due to the legacy of bad experiences with imposed cooperatives in the communist period. These results highlight the fundamental role of trust as a key value that can be enforced with repeated and positive interactions between people (the owners) and learning about the outcomes (the activities) to increase membership of PFOOs [47,70,73].

(C) Rules. PFOOs can be associations or cooperatives, both legal forms characterized by limited liability and democratic governance structures [65]. Cooperatives are enterprises typically characterized by the principle "one man, one vote", independently of the forest area owned [11], while associations are not enterprises. They may be nonprofit actors primarily acting as lobbyists and financed via membership [46] or enterprises where administrative and technical support is given to specialized professionals, eventually employed by the organizations, or purchased as consultants. In any case, no ownership rights are transferred to organizations, and forest owners democratically participate at some stage of decision-making [69] that could be only episodic (shareholder-type of governance) or continuous (cooperative-type of governance). Involvement of members in the governance structure also depends on their personal interest: active owners can be fully or partially

engaged in management activities but are surely part of the decision-making process, while "absent owners", those who live far away and have no contact with their forest property and are only interested in the forest as a family asset, also delegate to organizations most of the decision making [53,66].

(D) Power and Resources. We can observe different distribution of forest management responsibilities, once again according to participation of members, as suggested by some studies, from which four major models can be identified [52,53,69]: (a) active owners fully engaged in their forest activities, predominantly oriented to timber harvesting, which is performed by each member, who also transports material to the industry, while the organization performs the arrangement of timber sales, measurement and quality assessment, and invoicing and payment, and ensures the contract-fixed price of wood; (b) almost the same as model (a), with the difference that transportation is also entrusted to contractors; (c) organizations of "multi-objective" owners, whose main source of income is not related to forestry, and they spend little time performing activities in their forests, while most activities are left to the PFOOs; and, finally, (d) is the case of "absent owners", those who live far away and have no contact with their forest property and appoint organizations to carry out management, sales, administrative tasks, and all activities. When owners are fully involved in the management of their forests, professional skills can be found between them, inside the organization, to carry out forest operations; more frequently, external foresters are designated as technically responsible for forest management, and forest operations are contracted to external entities. The French CNPF supports PFOOs and even individual owners with consulting and training to steer their forest management toward sustainability [74] and finally evaluates the forest management plans that are mandatory for forests bigger than 25 hectares, while other easier documents are sufficient to orient (sustainable) forest management in smaller forests [75]. In many cases, the constitution of forest owners' organizations is financially supported by public funds: national, regional, or eventually derived from the Rural Development Programs [9,47,69,73], and some articles report that many owners believe that their organizations will survive in the long term only if permanently financed by public funds [61,69]. On the other hand, they should not rely exclusively on public financial sources, but rather gain direct economic returns from forest management and simultaneously deliver and value different value-added services [46].

*3.3. Commonly Managed Forests*

Organizational aspects of community forests, community forestry, and community-based forest enterprises are cited in 24 articles. Although there is some overlap, substantial differences, such as the allocation of forest property rights and the purpose, suggest separating these categories.

3.3.1. Community Forests

CFs are not properly a specific organizational model, rather an ownership typology; the allocation of land property rights to the community generally leads to the formation of endogenous organizations [76] that could have various forms. Therefore, CFs do not present a single organizational model or a homogeneous group of organizations, but different models can be found in different countries and even in diverse regions of the same country, because common goods' (eventually called 'commons') management organizations have typically been established in the past and have a strong traditional legacy. Common ownership rights, management, and use of natural resources (in our case, forests) are the characterizing traits of this category.

(A) Actors. Common property is a third ownership option beyond the well-known forms, namely, private and public property. Many European forests are owned by communities, even if the overall area covers a small share in the total European forests (a bit more than 2% [56]) in various forms: from traditional rural commons dating back to premodern times, typically in Spain, Italy, France, Austria, Slovenia, and Romania, to relatively more recent community-owned or -managed forests, established, for example, in Sweden, the UK,

Czech Republic, Slovakia, Poland, and Hungary [1,2,77]. Members of traditional commons are typically local families, for example, in the Alps, who have inalienable and indivisible rights [48], while in more recent community forests, outsiders could eventually have access to the common property (a share of it), so that today Swedish forest commons, for example, are owned by people, companies, the church, and even the state. The access to the common is a crucial aspect, strongly related to inheritance and, in some cases, to the possibility of buying the farm/household on which the commons' share is based. In some community forests, in the last few decades, the original actors have gradually disappeared and are being replaced by new actors who can have different demands on the resource [78]. CFs often involve forms of collaboration with exogenous political and economic actors that can be found in the same local context [76]. As for private forest owners' associations, the importance of second-tier organizations is cited for CFs, too. These organizations work to represent members politically, but also share information and generate coordination, could pool resources, and provide capacity-building projects [79]. In addition, CFs can be a key actor in local networks, thanks to their ability to deal with the market and work with other players in their territory and within the value chain and cultivate strong partnerships with local governments. From the literature, the case of Mersey Forest emerged in this sense, described [33] as a community forest recently established in the UK to lead a network of local governments, government organizations, landowners, private companies, and the community in implementing landscape changes.

(B) Values and Discourses. Communities that own and manage local natural resources are organized first of all to regulate the use and management of common resources. However, CFs do not limit themselves to forest management practices alone, but incorporate a broader set of goals, often involving diverse local stakeholders, again presenting elements and characteristics from private, public, and nonprofit organizations. Communities that own forests maintain a decisive role in the stewardship of the rural area in which they are rooted [48]; they can successfully set other purposes such as landscape conservation and restoration or the preservation of biodiversity [33,78]. Multipurpose management capacity, together with the ability to work in partnership with other local actors, may allow CFs not only to achieve their primary objectives, but also to become a community-driven organization, as described again for the case of Mersey Forest in the UK [33]. Trust, reciprocity, solidarity, and information sharing are indicated as key values for CFs that create capital on a level with natural, physical, financial, human, and political capital, representing a powerful instrument for building these other forms of capital [48,77]. In a local context, the collective action tends to develop with higher levels of social capital, defined as shared knowledge, understanding, norms, rules, and expectations about the patterns of interactions that groups of individuals bring to a recurrent activity [80].

(C) Rules. Some CFs' organizations are shaped like private enterprises (collectively owned). Some others have different organizational models, also depending on special state laws that regulate them. CFs' enterprises can operate as associations, employee-owned businesses, cooperatives, indigenous enterprises, not-for-profit societies, and firms owned by towns and municipalities [81]. Common property is a model of resource management that creates rules for the use of common property resources, defining who is and who is not eligible to benefit from the use of these resources [82], therefore in some ways defining who are members of a community, for the purposes of resource management. These models underpin the notion of 'decentralization', or 'devolution', of forest rights in that they leave it up to forest-dependent communities to govern local forest resources in ways that protect resource utilization and sustenance for collective goals [80]. CFs' enterprises are characteristically hybrid organizations, integrating public and private interests, objectives, and organizational elements, from the governance structures to the generation and sharing of profits [32,81,83]. Internal governance involves a decision-making body (a board) elected by the members' assembly, which is responsible for the management and economy and for the collective goals monitored by the assembly [76,78].

(D) Power and Resources. Hybridity seems to bring some relevant organizational challenges to CFs' enterprises: how to meet hybrid goals in an international marketplace and ask them to participate, in some way, in a complex global business network. Relevant governance challenges have been detected, as community members are responsible for technical, business, and administrative decisions but could not be sufficiently trained or skilled, and the governance structure is not always adequately designed to gain lacking competences and capacities [81]. Some authors underline the importance of distinguishing the roles and responsibilities of the enterprise members, staff, and board members. Decision-making roles and power shall be distinguished between the board of directors and the administrative staff [84,85]. In the Swedish model for CFs, for example, the shareholders' assembly elects a board, which is responsible for the management and the economy, but also, according to the law, a professional forest manager must be contracted or employed and is directly responsible for the forest management [78].

### 3.3.2. Community Forestry and Community-Based Forest Enterprises

Articles presenting community forestry are all about experiences from the UK and clearly distinguish them from the previous category (community forests), also proposing a sharp definition for community-based forest enterprises (CBFEs).

(A) Actors. Community forestry is broadly defined as those situations where communities are involved in the governance, decision-making, or management of forest and forest resources and gain some benefit from them [49]. Some groups could eventually own or lease their forests, and others manage them in partnership with another organization, usually the landowner, through a management agreement. It is noteworthy that, though in the literature we found the locution "community forestry" to indicate these experiences, in the UK, communities organized for community forestry are called "woodland communities" or "woodland groups". This is also the name used by their two main second-tier organizations: the Community Woodland Association (in Scotland) and Llais y Goedwig (in Wales), which are self-organizing associations, initiated by the groups themselves for mutual support and to represent their interests to policy makers [86]. Community forestry can be further organized in enterprises, namely, "Community-based forest enterprises" (CBFEs), more closely defined as experiences in which community members are organized into a company to actively produce goods and services in response to market demands, generating income, social returns, and other assets benefitting those communities [50].

(B) Values and Discourses. The main purpose is typically to produce direct or indirect benefit for a community through the management of forest resources. Enterprise and trading are not always primary objectives [49], though there are CBFEs strongly relying on trading. The aim may be to maximize profits to generate funds for the communitI(C) Rules. The major difference with CFs, described in the former subsection, lies in the fact that land ownership is not a prerequisite in community forestry and in CBFEs, since they can be carried out also contracting with private or public forest owners. Community forestry can be organized as community interest companies, cooperatives, or companies limited by guarantee (which are the options for CBFEs), but also as unincorporated associations and charities [32,81,87]. Communities are involved in the governance of forest land and directly in decision-making bodies, and they are always the first beneficiary, whatever the legal form of the CBFEs, that can be both for profit of nonprofit. Liability can change, according to the legal form, from a personal obligation of members in the case of unincorporated associations to limited liability in cooperatives and companies limited by guarantee. Consequently, a very broad set of governance arrangements can be found with very different degrees of community involvement. Decision making in companies is performed by directors or trustees, or by named post holders in unincorporated groups [87].

(D) Power and Resources. Forest management can be carried out by communities, as for the case of CBFEs, or contracted to third-party enterprises, securing them time-bound legal rights that may even exclude community use of woodland, which is the case of community-governed concessions, a relatively emergent typology. Community

forestry financing can be based upon trading, upon contracting with third parties who pay leases, or can be significantly based on grants, as for the case of charities and most social enterprises [32,85,87]. The positive impact of community forestry, typically focused on producing public benefits for the community, e.g., conservation and landscape values, allows us to look at this management solution as a viable option to realize the potential of forests in sustainable development [81,87], and growing case-based evidence can be found that community forestry delivers public benefits at a local scale and improves the sustainability of forest resources around the world [88].

*3.4. Social Forest Enterprises*

Although not yet a universally framed concept, social enterprises are growing in Europe and can be defined as entrepreneurial activities that do not trade for profit but are rather established for a social or environmental purpose; however, significant differences are reported between laws of the countries. SFEs are cited among six articles, and their best description was found in articles settled in the UK, where evidently there is a stronger tradition for this kind of organization involved in forest management. SFEs can be community-based forest enterprises and may also be chosen as an organizational model for CFs; anyway, they shall not be confused with the two former categories nor with the more general concepts of community forestry or nonprofit organizations. SFEs could not have a specific correspondence with a community and do not necessarily involve forest owners as members.

(A) Actors. Social entrepreneurs can be individuals, groups of people, or eventually entire communities when social enterprises are established for community forestry or to manage community forests. Social innovation can be a process for the creation of SFEs, defined as the "reconfiguring of social practices, in response to societal challenges, which seeks to enhance outcomes on societal well-being and necessarily includes the engagement of civil society actors" [89]. This definition suggests the key role that "civil society actors" have that can be easily recognized in the figure of founders (individuals or groups) who start the enterprise and frequently continue to lead it, in collaboration with family, friends, and trustees. The substantial work of volunteers is decisive in SFEs, but cooperation with other organizations is also an important factor, in particular with other social enterprises in the first region, that can be the SFEs' network of clients and providers [32,86].

(B) Values and Discourses. SFEs are often "hybrid organizations", since they try to combine the goals and cultures of both for-profit and nonprofit businesses; in forest-based SFEs, a triple dimension of hybridity can be seen, merging social, environmental, and financial goals, which can complement or compete with each other [86]. In any case, the purposes of the company are sustained by the key values of its members, which are typically solidarity, trust, care, and cooperation [51,81]. SFEs' value propositions can be based on trading forest products along the value chain, while in other cases, they mainly offer forest-based services, such as forest education and training, sustainable tourism, and recreation. Therefore, the forest can represent only a therapeutic setting or a venue for events. Another group of forest-based SFEs offers forest management services to other landowners, such as forest management consultancy or timber harvesting, while some enterprises developed a mixed income strategy [32,51,87].

(C) Rules. The UK laws define social enterprise by the purpose of a business, with primarily social objectives, that can be carried out by unincorporated and various incorporated forms of social enterprise, which include limited companies, community interest companies, industrial and provident societies, and limited liability partnerships. They are nonprofit organizations, meaning that surplus is reinvested into the enterprise, to maximize social and environmental objectives, rather than providing returns to owners and shareholders. SFEs can be built based on community engagement [32], which is the case when this model is adopted within community forests and forestry, but they could also be independent of the community and do not necessarily include it in governance or woodland management. Forests, the main asset for forest-based SFEs, can be held by third parties and contracted or

made accessible by the owner [87]. Specific governance structures are often used to manage the forest resource and business, the ethics and livelihood choices inherent in the business, and the integrated way in which they cooperate with other organizations within the region. Moreover, in addition to their social and environmental goals, they must combine with a set of business-like financial and managerial systems to meet their commercial objectives that are needed to cover their operational costs [32,51]. Managers of SFEs must consider all of these diverse goals and conduct their business on a challenging multiple-objective basis, considering the multiple interests of different stakeholders involved (such as participants, staff members, funders, partner organizations) while balancing the social, economic, and environmental dimensions of the SFEs.

(D) Power and Resources. Some SFEs depend to a considerable extent on volunteer work and grants [32], but others developed sufficient commercial activity sufficient to be financially independent. However, financial security is often reported to be the external factor that causes SFEs to crumble, also because funding mechanisms appear to have fallen short of fully assessing their performance, with long-term social and environmental effects largely neglected, so that monetizing ecosystem services provided by SFEs remains difficult, while related costs are tangible [32,51]. SFEs with strong asset ownership (forests) have access to a wider variety of income sources and can use land as capital against which to raise loans [87]. Although challenged by insecure financial performance, an increasing connection with rural development is reported for forest-based SFEs [51].

*3.5. Forestry Enterprises*

Another set of articles (n = 16) deals with "forestry enterprises", "forestry companies", "forestry contractors", "forest harvesters", or "forest workers cooperatives". General conclusions could be misleading when talking about this category, being quite broad in terms of possible legal forms and characteristics that forestry enterprises (FEs) can assume. No matter the name or legal form, "forestry enterprises" are forest workers' organizations whose business is based on forest management operations, contracted with either public or private forest owners.

(A) Actors. Members of FEs are forest workers, such as timber loggers and forestry machinery operators. Until some decades ago, forestry workers were mostly employed by big forest companies, where existing, or by the state and local administrations. Many harvesting enterprises began their activity in northern Europe (and in North America) when large-scale forest companies, starting from the 1980s and early 1990s, decided to outsource most of their harvesting operations, often offering to sell their machinery to selected machine operators who could then continue to work as independent contractors [64,90,91]. Similarly, some years later, in some European countries, such as Slovenia, Finland, and Baltic republics, SFMOs also started to outsource harvesting, transport, and reforestation [58,90]. In Finland, contractors' size can determine their "position" in the network of forestry operations: the largest companies often act as prime contractors for industrial buyers and then use subcontractors to perform some of the work, which seems to be a profitable strategy [92]. In countries with strong forest industries, FEs are often considered by other stakeholders as an extension of their clients' operations and, in some cases, this is also their own self-perception [93].

(B) Values and Discourses. The original purpose of this category was somehow inspired by third parties that pushed for the development of FEs: large-scale forest companies in the 1980s and early 1990s, and, later, public forest owners and managers outsourced most of their harvesting operations, as a consequence of reform processes aimed basically at improving efficiency by reducing costs and improving the productivity of forest operations, but also to gain greater capacity flexibility and reduce the bounded capital in expensive machinery [58,90,91]. Although efficiency was gained by large forest companies and state organizations, a general issue of low profitability afflicts FEs. Some of these enterprises react by innovating their business model, starting from the value proposition, as they begin to carry out other activities complementary to forestry, such as land maintenance works,

tree climbing, transport for third parties, or high-value and small-scale timber processing [94,95]. According to some studies, the successful business strategy of FEs is based on increasing knowledge through learning orientation, enabling continuous understanding of the surrounding environment and the attitude of innovation, together with strengthening organizational capacity, which is specifically referred to as the effort to operate in the most rational way as to reduce costs [96,97]. Interestingly, a study revealed that FEs exhibit a "clan corporate culture", which can be summarized with the use of team thinking, the implementation of individual development programs for employees, and the focus on creating a friendly work environment [98]. Cooperative FEs are based on further values such as mutual help, self-responsibility, democracy, equality (one member-one vote principle), equity, and solidarity and can therefore be inclined to other ethical values of honesty, openness, social responsibility, and caring for others [68].

(C) Rules. FEs can normally be described as micro and small–medium enterprises (MMEs and SMEs). These categories, introduced by EU recommendation 2003/361, are broad: many different legal forms can belong to them, and FEs can be companies with limited or full liability of the owners. FEs can be organized as cooperatives, a model characterized by involving the workers as members, i.e., simultaneously owners, controllers, and economic participants of the enterprise, whose activity is conducted with a prominent mutualistic scope [99,100]. Cooperative FEs have more structured internal governance, with decision-making authority assigned to a board, or eventually delegated to a CEO, and operational roles for other workers, whereas smaller FEs seem to have a simplified governance structure, where the owner(s) could be at the same time a worker and the team leader in the field.

(D) Power and Resources. FEs' performance is mainly oriented toward productivity improvement and technical and operational efficiency to achieve cost reduction. Small FEs must struggle with low profitability, originating from the frequent use of tendering by their customers, which creates tough price competition, especially because each contractor's radius of operations is limited to a few customers in the region [91,92], but also because they have limited power to negotiate for favorable contract terms and worksites with the large forest companies. Harvesting companies' activities are subject to weather conditions, strive for high investment costs for machinery, and often have limited internal business skills [64,100,101]. However, other authors describe FEs with high adaptation capacity, due to learning orientation and organizational capacity, that allow them to precede competitors with new ideas and encourage business development and diversification, also thanks to adequate structures, capital, and skills to carry out activities complementary to forestry [95–97].

Looking beyond their primary profit goal, from a more general socio-economic perspective, especially in disadvantaged rural areas, small forest enterprises can play a key role in the development of multiple dimensions of economic, environmental, and social prosperity at the local level [102], they significantly contribute to guaranteeing employment and managing land with positive environmental effects, including hydrogeological protection, biodiversity, and carbon storage [96].

### 3.6. Other Organizations

Some other organizational categories were detected within the 66 articles; however, it was impossible to complete their description, following the analytical framework, because of too scarce data reported in those papers, where they were just cited without deepening their characteristics. In this subsection, some information extracted from the reviewed articles on those other categories are reported.

ENGOs were found in two articles, focusing on their role and organizational adaptation, following changes in forest governance and policy. The role is recognized in participatory processes, established to address the diversity of interests among forest stakeholders that increased as the forest management objectives expanded in the last twenty years (at least), with the implementation of the sustainable forest management concept [103]. ENGOs

typically have a role in forest management as key stakeholders that can challenge forest managers and policy makers; however, they can also assume a direct role in forest management, being designated by forest owners (private, normally) to carry on management projects typically oriented to nature protection. Interest groups such as ENGOs developed a multilevel structure to improve democracy while increasing their ability to face multilevel governance characterizing the European forest sector, eventually structuring federations or participating in umbrella organizations [104].

Model forests is another organizational model, described in one article, mainly characterized by a governance arrangement that, associating a broad range of stakeholders among which consensus is established, works to ensure the sustainable development of the community on a territory characterized by forests, where forest management is carried out with highly participative decision-making processes. The organizational aspect is seen as the formation of mechanisms for sustainable forest management and for the improvement of the forest planning system, combining knowledge, resources, and experience for research in the field of forestry, introducing new methods of balanced forest management, and taking into account their own and public interests and features of a particular region [105]. Innovative organizational and business models are described in a study, in Austria and the UK, where very small, even one-man, companies develop new forest-related offers, mostly based on NWFP, which are sold not for their sole utility, but as carriers for an experience which is demanded by the customers. These businesses succeed by riding the wave of new interest in personal interaction in the use of NWFPs and reveal new opportunities and ways of using goods coming from the forest [106], embedded in very flexible and intersectoral organizational and business models, basically relying upon contracts between these small entities and other actors (i.e., public authorities, other local organizations).

Although certification schemes, such as FSC and PEFC, have a consolidated and unquestionable role in forest management in Europe, surprisingly, none of the 66 articles mentioned them clearly focusing on organizational characteristics related to them or to organizations certified according to their standards, apart from a work in which the arrangement for forest certification groups was described. In Lithuania, the "Group certification manager" was legally recognized as a non-profit nongovernment organization under the Law of Public Institutions, promoted by five wood processing firms that needed certified timber. Today, more than 180 individuals and legal persons have joined the group, all representing 90,000 ha of managed private forestland. The manager, who is not allowed to participate in any political debate related to forests, is appointed for sustainable forest management, advising forest managers on the implementation of certification requirements and developing cooperation between PFOs themselves, forest managers, and industries [46].

## 4. Discussion

Organizational adaptation and development can be recognized within the literature as responses to some of the challenges that occurred in the forestry sector in the last decades, either as necessary evolutions in response to drastic changes or as strategic choices for innovation and growth. Some dynamics are more recognisable and described in the literature, such as the development of PFOOs following the privatization of forests in former Soviet republics and Balkan countries; the establishment and development of FEs following outsourcing of forest operations by large forestry companies, or the establishment of umbrella organizations, first at the national level, then at the European level, for policy influencing. Others followed different development paths: the development of CFs, CFEs, and CBFEs in the United Kingdom; the evolution and reorganization of ENGOs, adapting to changes in forest governance; the evolution of SFMOs toward multifunctional management models; the growth and diversification of FEs.

Organizations change over time, adapting to external changes and reshaping themselves to better suit new needs and purposes, through a process that is called organizational learning [11]. Powell and DiMaggio [34] theorized that isomorphism is the reason why organizations change, through normative, coercion, or mimetic processes. This work did

not explore these concepts and dynamics, but they are emerging from the literature review as fundamental aspects to be considered and further investigated.

A multitude of organizational types (names) were observed within the literature for forest management organizations. After a deeper analysis of their characteristics, a categorization has been proposed based mainly on the identification of the members and on the relationship of the organization with forest owners, and also subsequently on the purpose (e.g., to distinguish some SEs from FEs). Finally, at least three "axes" emerge to qualify the actors and the purpose on which a categorization can be based, apparently dichotomously:

- The legal nature of actors, with two relevant sub-dimensions:
  - The distinction between public, private, and third sector (private, but oriented to public utility);
  - The distinction between legally recognized 'formal' organizations and informal organizations which have no legal recognition (e.g., households, certification groups).
- The relationship with forest owners, which may be internal to the organization (members) or external (partner/client/contractor);
- The purpose, between the profit/not-for-profit dichotomy.

It is difficult to establish a priority among these criteria for a categorization; rather, it seems useful to emphasize the importance of considering them all, at least to correctly characterize the categories identified.

As anticipated in Section 2.2.2, some more organizational 'typologies' were detected within the 66 articles, but the results were too poor to allow a complete analysis to describe their organizational model according to the framework and to present one or more additional categories. The proposed categorization is far from being a complete representation of the organizational models for forest management organizations in Europe, missing some surely relevant typologies such as those cited in Section 3.6, namely, ENGOs, Model Forests, and certification groups, and probably some more that did not even result within the literature review. Another missing category could be defined as "umbrella organizations", but that includes quite a variety of organizations. Some examples were cited within the articles and reported in Section 3, when related to the analyzed categories; however, it could be worth recognizing them as a category, encompassing umbrella organizations that connect forest owners/managers and other local forest organizations for supporting the members in relation to their general interests (as forest owners: CEPF, EUSTAFOR, USSE, FECOF, ...); or in specific fields of policy action such as certification (FSC, PEFC, Plockhugget, Naturland, ...); research and innovation policy (Forest-based Sector Technology Platform, Innovawood, European Wood Policy Platform); or environmental protection (FERN, Forest Movement Europe), (Taiga Rescue Network, Association Internationale Forêts Méditerranéennes, ...).

Even the choice of the categories proposed was surely determined (and limited) by the results of the semi-systematic literature review, and some shortcomings were accepted in this work. The SFEs category, for example, is very specific, recognized within six articles, presenting enterprises legally recognized as nonprofit, operating forest management, and established for social or/and environmental purposes. However, in this review, no mention was found for "B-corps", a typology that is growingly interesting also for the forest sector, which would share the same characteristics of SFEs, apart from not being necessarily nonprofit, since also for-profit companies can obtain the certification 'B-corp'. Similarly, some environmental organizations could assume forest management responsibilities, therefore being very close to the SFE concept, but they are not enterprises; hence, another category for nonentrepreneurial forest management organizations should be recognized, but none of them was detected within this review.

However, the objective of this categorization was to test the application of the conceptual framework of "organizational model" to propose it for a comprehensive representation of an organizational arrangement, rather than complete a full assessment of all the organi-

zations involved in forest management throughout Europe. Therefore, some considerations on the framework, detailed per each of its four (plus one) key dimensions, are displayed.

### 4.1. Actors

Within the four key internal analytical dimensions of the conceptual framework, "actors" have a central position. The six categories of forestry-related organizations were identified according first of all to two main criteria: who the members are and who the forest owners are, the two being sometimes coincident, as for the case of SFMOs, PFOOs, and CFs, and otherwise being separate, as for CBFEs, SEs, and FEs. These two variables are independent in the framework, while most of the others depend on members' choices, apart from laws, that are determined by the external context. Forest ownership is a major matter of concern in the field [1,2,47,56], and a first distinction is due between public, private, and collective actors. Beyond the motivation of the members, which was identified as a key feature already in the first part of this research, forest owners' attitude toward the organization is also another important trait, sometimes separated from the former. Forest owners can assume different roles and have an important influence depending on their direct participation in the organization or not and on their interest toward their forest property and management. These latter range from active owners fully engaged in forest management and operations to owners participating only in the organization's governance, to absent owners only interested in holding their property rights, eventually earning some profits deriving from a delegated management. External networks and partnerships, formalized or not, are also frequently indicated as a very relevant variable regarding "actors", and are in some cases a critical one to achieve the organization goals, thus influencing an organization's power, such as the case of "umbrella" (or second-tier) organizations of PFOOs aiming to influence policy making. Clearly, for organizations establishing external business activities, i.e., selling products and services, clients assume a key role. Communication, which is closely related to organizational values and discourses, is a key feature in any case to empower a selling strategy, to achieve educational objectives [45], and to improve reputation [5], therefore also influencing organizational power, especially with respect to external actors. Nevertheless, only a few articles focused on these two features that result quite neglected: this could be misleading, especially considering harsh conflicts often rising around forest management activities that could be better addressed with proper communication strategies.

### 4.2. Values and Discourses

Variables describing this key dimension are extensively detected in the literature. The reasons why an organization exists and its members work together, are often highlighted as a key aspect and represented the third criterion to establish the categories proposed, the first and second being, respectively, the identification of members and forest ownership. The ultimate purpose of an organization can be categorized between profit or nonprofit; however, such a sharp and simple definition misses the relevant research for multiple purposes that are typical, for example, of CFs and SEs, but also of SFMOs [5,45,51,81,83]. The purpose of organizations is based on the personal values of their members (of the founders at least) and participating in the effort for their achievement is a fundamental choice of individual members, resulting in a value proposition for the organization's clients and stakeholders.

"Internal key values" are a distinctive variable (grouped into the "discourses" dimension) established by the organization, which are very important for PFOOs [52,53,69,73], CFs [17,48], and SEs [51,81]. These values have a strong influence on internal informal rules, another key variable that is very important for CFs and CBFEs, but also in SEs, i.e., those categories evidently characterized by decision-making strongly reliant on values.

Business strategy is another characteristic emerging from the analysis, regarding not only the products and services offered (the value proposition), but also about the organization's decisions to improve performance or for developing the business [46,58,96].

It seems appropriate to add this variable to the framework, belonging to the dimension "values and discourses", but also to the "power and resources" (since it is intentionally determined by the decision makers of the organization) and "actors" (which are typically client-oriented) key dimensions.

*4.3. Rules*

Most of the articles simply describe the specific organizational subject mainly referring to a legal entity, according to specific national laws, somehow assuming that the legal definition is implicitly and completely representing the whole arrangement and it is enough to explain everything of an organizational model. Laws define how organizations can acquire a legal status and partially regulate the organizational process. The legal framework unquestionably influences the organizational processes of formal organizations, starting from their constitutive operating decision and lasting throughout their life, demanding compliance with several general and specific norms. Nevertheless, it also indirectly influences informal organizations, for instance, because they are not allowed to conduct what is established to be prerogative of legally recognized entities. The legal perspective may bring about a precise identification of some key features of an organization, such as liability for the entity itself and for its members. However, the sole legal definition seems an insufficient criterion to define and describe organizations since it does not represent who the key actors are, but rather only partially indicates what is the purpose, how it works, and why it exists. Despite these gaps, the formal identification of an entity, its property rights, and assets, which are typically legal features, is essential also for organizations in the field of forest resources management, where informal entities are relevant, though. In Europe, especially in some countries, there are huge forests managed only at a household level, while others are abandoned by owners, in forest contexts characterized by fragmented and reduced size, which cannot be properly managed with an entrepreneurial approach [62]. This basic problem is often addressed with normative initiatives for the adoption of organizational models that group small owners to encourage more organized and effective management [46,53].

However, variables related to the dimension "rules" are not only those defining the mere legal form, but they also frequently refer to governance structures, often meaning internal governance arrangements. These are the "means by which to infuse order, thereby to mitigate conflict and realize mutual gains" [107], to define decision-making processes and roles and distribute power and responsibilities [79] in a continuum of solutions between hierarchical and democratic governance structures, thus originating vertical vs. horizontal distribution of responsibilities. Formalized internal rules and regulations can be a very important feature, as reported for CFs [48,80], including disincentives and coercion (sanctions) as enforcement tools; meanwhile, positive incentives can be effective tools to motivate people and organizations, as an alternative to hierarchies [83].

*4.4. Power and Resources*

Forest management responsibility emerges in all categories analyzed as a key variable influencing the power distribution, that is, first, who is responsible for planning and second, who is designated to carry out forest operations. This specific feature was not included in the initial framework, but looking at the results it seems appropriate to integrate within the key dimension of "power and resources". Financial sources (and performances), even if it could seem obvious, is cited as a critical feature in many articles [9,45,58,85,93,96], basically influencing the possibility to carry out actions and operations. Similarly, the structure is a key aspect, highlighted within all the categories analyzed, most of which are struggling with cost reduction as a primary strategy. A dichotomous representation of the "costs structure" as either capital- or labor-intensive organizations results from the literature on forest management organizations. Incentives are frequently mentioned as fundamental tools to support forest management organizations, sometimes even as triggering factors for the formation of some associations, as in the case of PFOOs [47,61]. With the exception of a work specifically analyzing power in forest governance [39], the

three constitutive elements of power proposed within the ACP approach, namely, coercion, (dis)incentives, and dominant information, were rarely cited, revealing a scarcity of studies on power dynamics, both in terms of internal organizational dynamics, and with respect to interactions between forest management organizations.

Decision-making power, in contrast, was a frequently reported feature, mostly related to internal governance structures, that is, the assignment of roles and responsibilities, in articles discussing PFOOs, where sharing (or not) decision-making power with forest owners is a key choice that shapes the organizational model [11,69]. In CFs, the allocation of decision-making power is a constitutive trait to empower communities that recognize their right to administer their land (forests) administration [48,76,81,87].

*4.5. Other Key Variables*

Many articles discuss (or even cite) business models as a key topic for forest management organizations, intending to describe "the value a company offers to one or several segments of customers and of the architecture of the firm and its network of partners to create marketing, and deliver this value and relationship capital, to generate profitable and sustainable revenue streams" [25]. The business model is a separate concept from "organizational model", though it is based on organizational choices and, according to the broad conceptualization proposed, it could be considered a part of the whole, with respect to the concept of the "organizational model". The business model is a representation from another perspective; however, its variables describe what an organization does (the value proposition, nested in the "discourses" dimension of the framework), addressing who (the clients and beneficiaries, which is a characterizing variable within the "actors" domain) and by which means (these latter refer to the key dimension of "power and resources"), finally focusing on revenues and costs, which in turn are a measure of output and inputs of an organization's activity. It should be noted that considering the mixed economic nature of forest ecosystem services, encompassing many nonmarketable services, forest management organizations' business should not only be analyzed as "traditional" capitalistic businesses (that is certainly the case of many FEs). The business model concept should also be extended to social business and civil society-oriented business, where the word "business" is brought somewhere further from its traditional semantic domain, as in the case, once again, of some community forests and of some innovative forest-based activities carried out, for example, by charities, SEs, and CBFEs in the UK [32,33,51] but also by SFMOs [5,45,54]. The wider perspective of the organizational model seems to better capture this complexity, where an organization's business is built on the values and choices, not only upon a mere profit purpose, but more research should probably be conducted on this.

As hypothesized in the conceptualization, organizational forms are very sensitive to their context and coevolve with changes in their environment [108]. Therefore, the literature review confirms that the conceptualization of the "organizational model" must be framed in a larger (and even more complex) dimension, which is the context: legislation and governance, social norms, and other actors and relationships, but also natural resources, global, and local environmental issues [109]. Into this frame, accepting a very general simplification, at least two more variables should be added, to describe this fifth key dimension: influences and impacts. In addition to the influence of context on organizations, their activities have an impact on the context: positive and negative impacts of forest management on the context can be recognized as another key variable of the framework. The capacity to provide multiple forest ecosystem services, together with the main value proposition, is counterposed to models that achieve one or a few ecosystem services, ultimately limiting the provision of some others. We can observe a differentiation between the identified categories: SFMOs manage forests to maximize many ecosystem services, that is, conservation of nature, protection of water and soil, cultural services, together with provisioning services [5,45]. Landscape conservation and cultural services are often management objectives for CFs [17,48,81] that frequently have positive social impacts, the latter explicitly addressed by SFEs [51,81], but cited also for CBFEs and FEs [87,96]. Despite

financial challenges, community-based (CFs and CBFEs) and social-oriented (SFEs) models are reported to be definitely promising organizational solutions to manage and govern natural resources in ways that improve the lives of local communities and promote resource conservation, bridging forests (ecosystem services provider) and society (ecosystem services receiver), thus expected to sustain a broad set of forest ecosystem services [51,81].

Ultimately, we identified five more variables that might be relevant for a comprehensive analysis of organizational models (Figure 3). The complete framework is represented in Figure 3, where organizational variables are assigned to each of the four inter-related key dimensions, framed into the context.

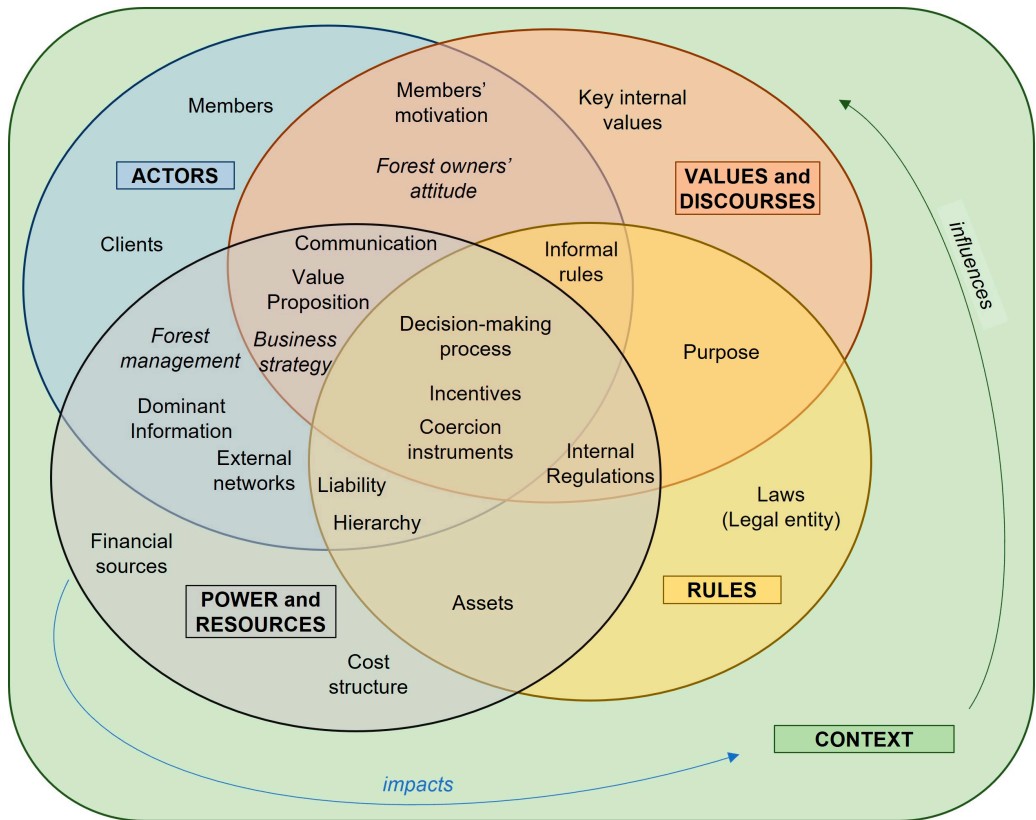

**Figure 3.** A refined conceptual framework to represent and analyze organizational models in the forestry sector (inspired by Arts et al. [36]).

## 5. Conclusions and Recommendations

As expected, no uniform conceptualization of the "organizational model" was found in relation to the forest management domain in Europe. This paper is an attempt to clarify the foundations while embracing the complexity of organizational arrangements in this specific domain. "Organizational model" is conceptualized as a representation of the way one or more "actors" establish internal and external relationships, set order ("rules"), manage responsibilities ("power and resources"), to achieve their purpose ("discourses"), influenced by a "context" that, in turn, is impacted by their activity. Twenty-five variables were used to describe the various and diverse organizational models within European forest management organizations. Despite the fact that the word "model" could suggest the search for a replicable representation, complexity is the major trait emerging from this conceptualization, so generalization should be avoided. Organizations are complex entities, and considering them under a single perspective (e.g., the legal aspects or the business model) could be misleading if not acknowledging this incompleteness. It seems more appropriate to encourage a holistic approach, where the ability to assess, to develop, and to harmonize the multiple dimensions is the priority, rather than directly incentivizing the establishment or the replication of apparently successful organizational types and

business models. A more open approach could also allow the recognition of innovation opportunities hidden within informal organizations.

Finally, the categorization proposed is far from being a complete representation of the organizational models for forest management organizations in Europe; however, this analysis enabled an overlook of many different organizations, often indicated in the literature with different names, providing some (about twenty) detailed characteristics per each of them.

Many more topics related to organizational models were mentioned in the text, suggesting the opportunity for further research to be developed in this field. Meanwhile, some shortcomings of this research must be acknowledged: the first part dedicated to conceptualization is based on quite a rapid and general design relying only on some of the existing theories. Synthesizing from all available theories, to structure a more solid new organizational theory was not the scope of this work; therefore, our conceptualization is built just on portions of some theories. However, the semi-systematic approach was chosen to review the literature, whereas a full systematic review could better suit the goal of evaluating and classifying all the forest management organizations. Again, this was not the purpose of this research; rather, it is dedicated to proposing an attempt of comprehensive conceptualization, suitable to draw a characterization of forest management organizations, and other methodologies could improve both the conceptualization and the categorization.

**Supplementary Materials:** The following supporting information can be downloaded at: https://www.mdpi.com/article/10.3390/f14050905/s1, Table S1. Searches and results for the semisystematic Literature review; Table S2. List of eligible articles selected at the end of the semisystematic literature review process; Table S3. Categorization and analysis of organizational models identified through a semisystematic literature review in the European forestry sector.

**Author Contributions:** Conceptualization: F.L., L.S. and D.P.; Methodology: F.L. and L.S.; Formal analysis and Data curation: F.L.; Writing—draft: F.L. and L.S.; Writing—review: F.L., L.S. and D.P.; Visualization: F.L.; Supervision: L.S. and D.P. All authors have read and agreed to the published version of the manuscript.

**Funding:** This research was developed within the PhD program at the LERH (Land, Environment, Resources, and Health) School, financed by the University of Padova, and did not receive external funding.

**Data Availability Statement:** Not applicable.

**Conflicts of Interest:** The authors declare they have no conflict of interest.

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
