# Peer review of "Organizational Models in European Forestry: An Attempt of Conceptualization and Categorization"

_forests, doi:10.3390/f14050905_

Round 1

Reviewer 1 Report

The authors present a relevant research topic: aiming to categorize and better describe forestry organizations in Europe.

There are, however, a number of substantial concerns so that I recommend to accept with major revisions

In general:

-        Make the theoretical model (Tab.1) more comprehensive and „dynamic“ in a sense that key features can be assigned to several dimensions (or provide a more conclusive assignements to the key features). Acknowledge the real authors of the key dimensions – this is not your academic work – what you did is to assign the features – and these are in parts debatable and not complete.

-        The focus is very much on economic organizations, governments and forest owners. Environmental organizations are hardly appearing. Why?

-        Provide a subchapter on the analytical approach: how is the theoretical framework linked to the analysis?

-        How did the analysis work? Describe the meta-ethnographic approach better

-        Sparate results from discussion. This would help to better understand what you did – if you present your results separately.

-        Ask a native speaker/editorial writer to revise. I gave some examples of typos and linguistic drawbacks below.

Some details:

41-42

Already here the focus is set very narrow: governments and companies

91

„How (does the work …)““ instead of „How (do the work …)“

114

„persons“ instead of „persons‘ “

116

„enters“ instead of „enter“

140

„does“ instead of „do“

159

„allows“ instead of „allow“

162

Section 2.1.3 is particularly nicely written, I enjoyed reading

181 - 199

In fact the key dimensions are a repetition of the political concept from the literature you cite and you are just applying Arts et al (your citation no. 31) where it says in the abstract „Policy arrangements refer to the substance and the organisation of policy domains in terms of policy discourses, coalitions, rules of the game and resources.“ – you just need to replace Art’s „coalitions“ with your „actors“. Whats „PAA“ that seemd to have inspired the framing?

Tab 1

The four key dimensios are ok – as long as you do not sell this as own concept (see above) but e.g. as „modified based on Arts et al. The idea to assign key features is nice but the key features and disciplinary domain columns are somehow random and incomplete. Perhaps a matrix would be better than a table where one feature could be assigned to several dimensions? Some examples:

Actors: social sciences certainly also have networks as a key features, see all the work related to social network analysis where networks are THE main feature of actors

Why is communication not assigned to social sciences? In 2.1.2 communication is even mentioned (line 158) and to my understanding it is certainly a feature in social sciences.

Power is featured much too narrow - see Krott‘s actor centered power approach with information, incentive and coercion elements. In your Tab 1 power only is comprising economic features. But if you would widen your power perspective then certainly social networks (power is derived from cooperations) and political domains have power aspects. E.g. coercive power has often been interpreted as regulatory power by e.g. governments. You even mention power yourself under point (v) in 2.1.2 – social dimension.

Hierarchy certainly has to do with networks and actors as well

In „Laws and rules“ informal rules are missing – you cited Ostrom in 2.1.2

Why is decision making assigned to „Laws and rules“ only? - it certainly hast to do with actors, discourses and even with power if you take Krott‘s informational power aspect into account. He says that power means to alter the behaviour of someone else – this certainly has to do with a decision of this person …

203

I do not understand how you link your theoretical framework in Tab 1 to the literature search? Why do you set up a model (Tab. 1) if you do not apply it in the actual research activity? This is essential: a section explaining your analytical approach is missing. You present some theory only and then run a literature serach, but then: do you assign your findings into the theortical categories? How does the conceptualization of your reasearch work? How do you come from Tab 1 to Fig 2? Before you describe all the details in the results and discussion section you need to describe your „plan“.

208

„have“ instead of „has“

211 – 228

I do not understand, but honestly I am not an expert in programming literature searches, can you reformulate? If not, I „believe you“

213

„W/3“ ? – but perhaps this is a technical issue that I am not aware of

218

You searched for the words „hot topics“ ?

222

„resuls“?

240

„deletd“

242

„because not fully consistent“ – verb is missing

245 – 257

The language needs specific improvement. The explanation of the meta-ethnographic approach is not clear. I am sure in a para of 12 lines it is possible to explain what you did even to one who never applied this approach and does not know it.

249

„organizazional“

250

„forrestry“

271

„actors“ instead of „Actors“

Table 3

This has a high emphaszie on economic organizations and forest owners. Environmental organizations are mentioned, yes, but given the wealth of them in Europe they would certainly „deserve“ an own aggregate extra from „social“. Please consider: Sustainablility is defined as comprising economic, social and ecological aspects and the weight that your analyzed organizations gets should reflect this. I assume that your search is somehow biased towards the economic perspective. Either you exclude environmental organizations by definition, e.g. claiming that „forestry“ in the title of the article is a very traditional perception oriented towards economic benefits, or you have to give them appropriate weight.

258 – 911

It is really tiring to read because the reader does not understand whether you describe your findings or discuss them – this seems to be a complete mixture. Findings would be all results that help to structure, sort, group … forest organizations based on your theory and your literature search. Discussion is all the context information and valuation of your findings – this needs to be separated. I recommend the classical approach: first resutls then discussion.

908

Fig 2 is more or less a repetiton of Tab. 1 in graphical form. Is this now the main finding? How did you come from the original 16 to the 20 variables? And the minor changes that you do are not conclusive: now you insert „Forest owner’s attitude“ to Actors, but does this not have to do with the values (Discourses)?

916-918

No, this is not on the basis of your analysis you just repeat and apply what Arts et al have conceptualized already in 2006, see above.

Reviewer 2 Report

Dear Author/s,

thank you for very interesting paper, and relevant topic of different organisational forms in forestry.

Since I find this overview as very interesting and relevant for future research, I would suggest further working on it and possibly publish it after major revisions are done.

I have impression that paper brings in many different notions and concepts and try to mix them all, and in this way, it loses clarity and focus. I am really not sure if that is necessary, as just bringing all these information on different organisational forms and how are they tackled in the literature would be already interesting.

Some of the main downsides of the paper are following:

-        Much of the introduction, whole second paragraph and also part of the third, relate to the concept of innovations and different types of innovations and how are they important for forestry – however later in the analysis this aspect is not mentioned at all. I was expecting paper will talk how different organisation relate to innovation and in which way, but that is not the case. So, either rewrite introduction so it fits analysis or add aspects of innovation in the results and also discussion and conclusion section.

-        Similarly, “business model canvas”, business model thinking” and other aspects used in introduction are then not relevant for the result, where you cluster different organisational forms.

-        In the introduction you combine definitions of “organisation” and “model” and provide your own interpretation of what organisational model means… this is later related to the PAA approach in the approach section (section 2). This can maybe be added to methods, as it relates already on scoping your search and analysis. In this section 2 you already put three different perspectives for looking at the organisations, and what are similarities. This could be fitting better there.

-        However, the whole section 2 is very shallow, and it ends suddenly in using PAA approach, which is also the only one mentioned when discussing Policy sciences. I suppose there are many more approaches or definitions to organisation from policy sciences. So somehow is not very clear why you stick just to some concepts from these “scientific blocks” and not to others etc. Somehow, I would make it simpler and easier to follow, and check if it is necessary all this complexity, that is then anyhow entailed within PAA approach.

-        Regarding the PAA approach – even I find its use here as very superficial, covering each of these very complex 4 analytical blocks could be used as a guiding analytical element when reading/analysing literature. Even it is hard to dig from literature some aspects, especially Discourses and Power (that are both very complex phenomena that ask for much deeper analysis and use of more specific methods for data gathering). But if you mention that this is used just as a frame to cluster specific aspects and maybe also mention this in limitations of your research it can be justified.  

-        In the section 2.2. there is need to better explain key words, maybe adding it to annex. In the line 227 “some more keywords were defined” – which one? Please provide more detailed so anyone could repeat the search.

-        In result section you give table 2 where you again bring too many concepts which are according to you related to “organisational model”, and I think some of them goes in much different direction from what you cover in “organisational model”- Like Business model in some company or enterprise, can relate to one of the business strategies they use, and it can change form year to year or depending on the specific product in focus, which at the end is not related to organisational form or this company / enterprise. Also, Organisational governance is quite broader terms… So, I am a bit confused on how do you use all these later in the paper.

-        I find actually table 3 and then analysis of each of these groups as interesting. And I would suggest somehow simplifying paper and starting it from here.

-        Maybe you discuss limitations of using literature review as method n the discussion, and it would be interesting justifying use of this methods. And then why you use just Scopus as a search? What about using State of Europe Forests or UNECE report of private forest ownership as well as important source of information when clustering organisational types. At least to better describe “Actors” from different countries. I believe from existing literature you miss many information, but please explain strong and weak sides of the methods used.

-        When describing each of organisations, you say in hoe many articles these were covered. It would be interesting to see which papers were these. Maybe provide list of analysed papers in the annex.

-        In section 3.2.4. I would also appreciate more explanation. It seems that you equalise responsibility and power.  Maybe these summary sections in 3.2 are the place where you can bring in commonalities, as you did, but also identify shortcomings or limitations of what you could find in the literature.

-        And why do you have only here “other variables” and not in every single analysis. It is somehow abrupt how you again come here to business model concept.

-        In this Figure 2 and in the text where you describe model, In Actors you write “owners attitudes”, I think “attitudes” are not necessary in this category but rather they belong to “discourses”

-        In Conclusion you write that your “organizational model” brings broader sight… but I actually do not understand what is difference to “well-known business model” and it would be good stating what is new about it. I find your classification and description of main elements as something useful; it is nice reading about all different organisational forms in one paper and understanding complexity. The main strength is in synthetising it all. So maybe try to be modest with it, it is in any case interesting and relevant! It is about clysifing different organisational forms in forestry in Europe. I do not know why do you complicate it with introducing new concepts to it. Having these 4 categories and 20 variables is already enough to cluster them, and I find all other concepts you use as not needed and they complicate the paper.

-        One minor remark, in the first sentence of introduction (line 24 to 29), it would be good putting reference after each challenge, and not at the end as a group of references.

Round 2

Reviewer 2 Report

Dear Authors,

thank you for taking into a consideration most of the comments I made, and for improving your paper. I suggest it in this form for publication.